# Antimicrobial Resistance and Incidence of Integrons in *Aeromonas* Species Isolated from Diseased Freshwater Animals and Water Samples in Iran

**DOI:** 10.3390/antibiotics8040198

**Published:** 2019-10-28

**Authors:** Reza Ranjbar, Reza Salighehzadeh, Hassan Sharifiyazdi

**Affiliations:** 1Molecular Biology Research Center, Systems Biology and Poisonings Institute, Baqiyatallah University of Medical Sciences, Tehran 1435916471, Iran; 2Department of Clinical Sciences, School of Veterinary Medicine, Shiraz University, Shiraz 7144169155, Iran; sharifiy@shirazu.ac.ir

**Keywords:** *Aeromonas*, multidrug resistance, integrons, mreshwater animal

## Abstract

*Aeromonas* spp. is one of the major pathogens of freshwater animals. There has been little research on the genetics of antimicrobial resistance associated with it in Iranian aquaculture. To remedy this lack in research, 74 multi-drug-resistant *Aeromonas* spp. were isolated from farmed diseased carp, trout, sturgeon, ornamental fish, crayfish, and corresponding water samples and examined for genomic integron sequences. Class 1 integrons, containing seven types of integron cassette arrays (*dfrA1-aadA1*, *dfrA1-orfC*, *dfrA12-aadA2*, *dfrA12-orfF-aadA2*, *dfrA15*, *dfrB4-catB3-aadA1*, *aac(6’)-Ib-cr-arr3-dfrA27*) were found in 15% of the resistant isolates; no class 2 integrons were detected in any of the resistant isolates. As some tested isolates were resistant to more than two groups of antibiotics, our results demonstrated that freshwater animals in Iran could be a source of multiply drug-resistant *Aeromonas* spp. This finding suggests that the origin of the antimicrobial resistance of these animals be placed under increased surveillance in the future and that the use of antimicrobials be limited in aquaculture.

## 1. Introduction

*Aeromonas* is prevalent among freshwater animals. Motile aeromonads, including *A. hydrophila*, *A. veronii*, *A. caviae*, and *A. sobria*, are reported as facultative pathogens that can infect aquatic species, including fish, shrimp, reptiles, amphibians, and other species [1]. *Aeromonas* infections have caused large numbers of mortalities in recent decades, triggering enormous financial losses in the aquaculture industry. Moreover, warm-blooded animals, as well as humans, are susceptible to diseases due to *Aeromonas* via polluted water or food [1].

Because of these diseases, antimicrobial drugs are extensively used in animal husbandry and the aquaculture industry, and bacterial resistance to antibiotics is a harmful consequence of this practice. Clonal selection or gene transfer is the main cause of resistance in bacteria. Some genetic components, such as plasmids, integrons, and transposons, are partly responsible for spreading the determining factors of genetic resistance among bacteria [2]. Mobile integrons encoding integrases are involved in the recombination of gene cassettes and participate mainly in the dissemination of antimicrobial resistance genes [3]. A variety of Gram-negative bacteria has been detected and the most common integron is the class 1 integron [4]. It has been proven by some researchers that this integron in *Aeromonas* species is primarily type 1 and transports different resistant gene cassettes. Aminoglycoside resistance genes, i.e. *aadA1*, *aadA2*, and the trimethoprim resistance gene *dfrA1*, are the most prevalent resistance genes related to the class 1 integron [5].

Motile *Aeromonas* species are important pathogens in aquatic animals and have been given special attention. Some types of fluoroquinolones and sulfonamides are used in veterinary medicine and have been confirmed as a treatment for these diseases [5]. Therefore, antibiotic resistance in the aquaculture industry is widespread. Integron-inserted gene cassettes have been reported in *Aeromonas* species isolated from fish in different countries, including China [6,7], Thailand [8], USA [9], Australia [10], Mexico [11], and South Africa [12,13,14]. However, the prevalence of the class 1 integron in Iran’s aquaculture industry has not been addressed, and most studies in this area have focused on the prevalence of class 1 and 2 integrons in clinical isolates and environmental samples [15,16,17,18,19,20].

In this study, the distribution of integrons in genomic DNA from aeromonads isolated from farmed diseased freshwater animals in Iran has been investigated.

## 2. Materials and Methods

### 2.1. Bacterial Isolation 

Altogether, 74 *Aeromonas* species were isolated from freshwater animals suspected of having motile Aeromonas septicemia with signs of disease, including dermal ulceration, external and internal body hemorrhages, fin rot, lethargy, unbalanced swimming, and loss of appetite. The animal samples included 104 carp (*Cyprinus carpio*, *Ctenopharyngodon idella*, and *Silver carp*), 40 water samples from warm-water fish farms, 24 aquarium fish (*Carassius auratus*, *Heros severus*, *Herichthys cyanoguttatus*, and *Haplarchus psittacus*), 15 water samples from aquarium fish, 25 rainbow trout (*Oncorhynchus mykiss*), 15 water samples from cold-water fish farms, 5 freshwater crayfish (*Astacus leptodactylus*), and 20 sturgeon (*Huso huso*, *Acipenser stellatus*, and *Acipenser baerii*) from different farms of Fars, Khuzestan, Tehran, Mazandaran, and West Azerbaijan Provinces of Iran. The freshwater species were cultured as a protein source for human consumption. 

Samples from fish kidneys, body surfaces, and the collected water were aseptically cultured on brain heart infusion agar [21]. Typical colonies were isolated after 48 h at 25 °C and subcultured again. Subsequently, the bacterial colonies were exposed to Gram’s staining, cytochrome oxidase, catalase activity, and nitrate reduction tests for primary identification. Furthermore, Gram-negative, oxidase-positive, catalase-positive and catalase-negative, and nitrate reductive positive colonies were isolated. *Aeromonas* at the genus level and further key biochemical tests in API 20E (bioMérieux, Craponne, France), consisting of 21 tests that allow for differentiation among *Aeromonas* species, were performed following the manufacturer’s instructions [21]. Polymerase chain reaction (PCR) amplification of *16S rRNA elastase*, *lipase*, and *aerolysin* genes (Table 1) was carried out for further identification as described in previous studies [22,23,24,25].

### 2.2. Antibiotic Susceptibility Test

The resistance test was performed for all strains with 13 antimicrobials. The disk diffusion technique and commercial disk (Padtan Teb Co, Tehran, Iran) were applied for this procedure. These antibiotics included amoxicillin (10 μg), cefotaxime (30 μg), trimethoprim/sulfamethoxazole (1.25/23.75 μg), rifampin (5 μg), ciprofloxacin (5 μg), norfloxacin (10 μg), ofloxacin (5 μg), tetracycline (30 μg), doxycycline (30 μg), streptomycin (10 μg), gentamycin (15 μg), amikacin (30 μg), and chloramphenicol (30 μg) discs. The results were defined as susceptible (S), intermediate (I), or resistant (R) based on the interpretative principles from the Clinical and Laboratory Standard Institute (CLSI) [28]. The *Escherichia coli* ATCC 25922 reference strain was employed as the control [18].

### 2.3. PCR Detection of Integrons

The boiling method was applied for the extraction of the genomic DNA of the bacterial isolates [29]. Template DNA was stored at −20 °C until further use. The primer that was used for detecting the integrons, including class 1 and class 2, is presented in Table 1. The PCR procedures were done based on these phases: initial denaturation at 94 °C for 5 min, then 30 cycles of denaturation at 94 °C for 45 s, followed by annealing (60 °C for 1 min), extension (72 °C for 1 min), and a final extension at 72 °C for 5 min. Ten microliter samples of each PCR end-product were examined using 1.5% agarose gel electrophoresis and carried out with 1X Tris-acetate EDTA buffer (TAE) at 100 V for 1 h. A 100-bp ladder DNA molecular marker was included for each electrophoresis measurement (K-Plus DNA Ladder , SinaClon, Tehran, Iran). Afterward, RedSafe (Intron Biotechnology, Seongnam, South Korea) was used to stain the DNA bands, and they were next viewed using UV transillumination.

### 2.4. Amplification and Sequencing of Gene Cassette Regions

PCR method with primers shown in Table 1 was used for the amplification of the gene cassette arrays of *intI1*-positive strains. PCR products were sequenced and the DNA sequences were analyzed using BLAST (http://blast.ncbi.nlm.nih.gov/). 

### 2.5. Statistical Analysis

Susceptibility, data were compared using a chi-square test with SPSS software, version 18.0 (Chicago, IL, USA). Both susceptibility and resistance were calculated as percentages with 95% confidence intervals. A *p*-value < 0.05 was considered to be statistically significant.

## 3. Results

### 3.1. Identification of Aeromonas spp. from Different Aquatic Animals

Seventy-four *Aeromonas* isolates were identified according to genetic and biochemical features. All the isolates belonged to six species of *Aeromonas* (*A. hydrophila*, *A. veronii bv. sobria*, *A. bestiarum/piscicola*, *A. media*, *A. jandaei*, and *A. aquariorum*; Table 2), and the most common species was *A. hydrophila* (49/74, 66%).

### 3.2. The Antimicrobial Susceptibility of Aeromonas spp.

Approximately, 40.5% (30/74) of isolates were resistant to three or more classes of antibiotics. Isolates were resistant against streptomycin more than other classes (89.2%); data concerning the resistance rates of other isolates were as follows: ampicillin (85.1%), rifampin (78.4%), and vancomycin (70.3%). On the other hand, most isolates were susceptible to cefotaxime, chloramphenicol, ofloxacin, ciprofloxacin, and amikacin. The resistance rates for these antibiotics were 16.3, 22.8, 13.4, 18.4, and 13.4%, respectively. 

Susceptibility to antibiotics was different among the six detected *Aeromonas* spp. Greater degrees of resistance were observed in *A. hydrophila*, *A. media*, and *A. veronii bv. Sobria.* Regarding different hosts, the *Aeromonas* isolated from the sturgeon species was more resistant to trimethoprim/sulfamethoxazole, quinolones, tetracyclines, and streptomycin.

### 3.3. Detection and Characterization of Integron and Gene Cassettes

Among the 74 obtained isolates, *intI1* was detected in 11 (14.8%) strains, but *intI2* was not identified. Based on the statistical analysis performed, there was no significant difference among different bacterial species in the presence of integrons (*p* > 0.05). Moreover, the highest frequencies of the class I integron were related to sturgeon (n = 5), rainbow trout (n = 3), water samples (n = 2), and crayfish (n = 1). It should be mentioned that the class I integron was not found in the carp and aquarium fish isolates.

Seven types of gene cassette arrays were found among class I integron-positive isolates (Figure 1). In this study, *dfrA12-aadA2*, *dfrA12-orfF-aadA2*, *aac(6’)-Ib-cr-arr3-dfrA27*, *dfrB4-catB3-aadA1*, *dfrA15*, *dfrA1-aadA1*, and *dfrA1-orfC* were found (Table 3). Among the mentioned gene cassette arrays, trimethoprim-resistance genes were found in 100% of the arrays, and aminoglycoside-resistance genes were also detected in 90.9% of the arrays. Roughly 90.9% of the arrays encompassed both aminoglycoside- and trimethoprim-resistance genes in the same array. In brief, ten gene cassettes that were related to resistance against five classes of antibiotics were detected in the class I integrons. The genes were associated with resistance to aminoglycosides (*aadA1*, *aadA2*), trimethoprim (*dfrA1*, *dfrA12*, *dfrA15*, *dfrA27*, *dfrB4*), rifampicins (*arr3*), amphenicols (*catB3*), and quinolones (*aac(6′)-Ib–cr*). We also identified two open reading frames (*orfC* and *orfF*) with undiscovered functions. 

## 4. Discussion

In aquaculture, disease diagnoses are often presumptive, and disease treatment is done even though the data pertinent to antibacterial resistance is not reliable. Although there are just a few antimicrobials that are permissible by authorities in Iran’s aquaculture industry, the misuse of antimicrobials is prevalent in Iran. The emergence and dissemination of antimicrobial resistance among *Aeromonas* spp. from aquaculture has attracted much attention [6]. As we have found and other researchers have reported, resistance to a variety of antibiotics is confirmed in *Aeromonas* strains. Our data indicated that these resistance patterns in *Aeromonas* could be derived from the integrons encoding genes conferring resistance to trimethoprim-sulfamethoxazole, streptomycin, and other agents. Multi-drug resistance mediated by integrons in *Aeromonas* may not only complicate future antibiotic therapies used in aquaculture but also stimulate resistance gene transfer [6].

Integrons with gene cassettes are detected principally in the family Enterobacteriaceae, which is a Gram-negative bacterium. These are often situated on conjugative plasmids or transposons and can expedite lateral transfer between the pathogens [3].

The existence of integrons on similar inserted-gene cassettes in fish-pathogenic *Aeromonas* isolates with a frequency of 30–50% has been reported by several researchers [8,9,10]. Based on the results of the current research, the class 1 integron has was found in 14.8% of *Aeromonas* isolates from freshwater animals. Generally, in Iran, the frequency of class 1 integrons in *Aeromonas* isolates from cultured freshwater fish has been less than that of other freshwater species. This can be due to the different sources of isolates and the particular background of antimicrobial usage.

The *aadA1* and *addA2* genes are in charge of resistance to streptomycin while the *dfrA1* and *dfrA12* genes are responsible for resistance to trimethoprim. In this study, these genes were detected in most of the *Aeromonas* that carried integron. The most frequently resistant genes in the changeable region of integrons in different species of bacteria are *addA* and *dfr* genes [6]. The same model has been observed in *Aeromonas* isolated from aquaculture farms [8,30], environmental samples [31], and clinical isolates [32].

The *addA* and *dfr* gene cassettes are more dominant than other gene cassettes. This shows that these gene cassettes are more resistant when jointly compared with other gene cassettes. Moreover, in this study, the selection and dispersion of *addA* and *dfr* genes in integrons could be attributed to the large-scale use of antibiotics, such as streptomycin and trimethoprim-sulfamethoxazole, to control the diseases. 

In this research, a new cassette combination *aac(6’)-Ib-cr-arr3-dfrA27* in *A. hydrophila* isolated from a sturgeon fish was detected. This combination had previously been found in *Escherichia coli* from clinical isolates in China (GenBank EU675686.2), and *Staphylococcus xylosus* isolates from eel in China (GenBank KR259316.1). Also, a new cassette combination, *dfrB4-catB3-aadA1* was found in *A. hydrophila*, isolated from a crayfish. This combination had previously only been observed in *Aeromonas veronii*, isolated from Integrated Farms in China (KR067582.1). The *aac(6’)-Ib-cr*, *arr3*, and *catB3* genes responsible for resistance to fluoroquinolone, rifampin, and chloramphenicol were identified in this study, respectively.

In conclusion, the results of this study revealed that freshwater species could be a source of *Aeromonas* species containing integron-mediated multiple drug-resistant phenotypes. The cooperation of multi-drug resistance with integrons heightens the danger of co-selection and persistence of other resistance determinants under the selective pressure imposed by the use of antimicrobial agents. Therefore, the use of antimicrobials should be aimed toward advancing the healthy development of the aquaculture industry in Iran.

## Figures and Tables

**Figure 1 antibiotics-08-00198-f001:**
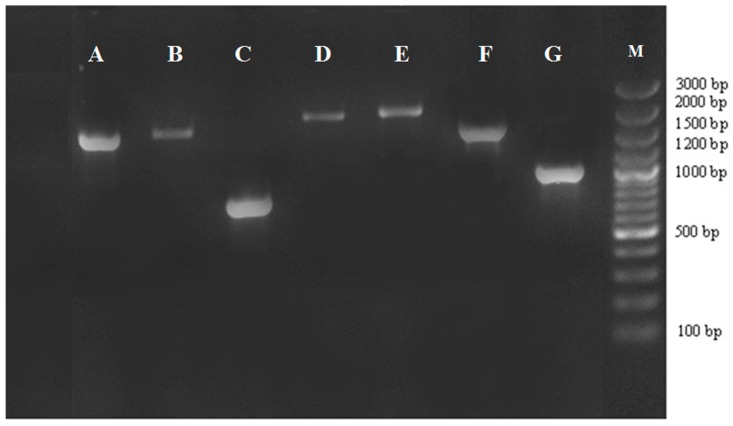
PCR amplification of gene cassettes in class 1 integrons. Bands with different sizes harbor different resistance genes. M: marker 100 bp; lane A–G: seven groups of class 1 integrons (A, *dfrA1-orfC*; B, *dfrA1-aadA1*; C, *dfrA15*; D, *dfrB4-catB3-aadA1*; E, *aac(6’)-Ib-cr-arr3-dfrA27*; F, *dfrA12-orfF-aadA2*; and G, *dfrA12-aadA2*).

**Table 1 antibiotics-08-00198-t001:** Primers used for the identification of *Aeromonas* species, integrons, and the variable region.

Target Gene	Primer Sequence 5’→3’	Size	Reference
*16S rDNA*	AGAGTTTGATCCTGGCTCAG ACGGCTACCTTGTTACGACTT	1500	[22]
*16S rDNA*	GAAAGGTTGATGCCTAATACGTA CGTGCTGGCAACAAAGGACAG	685	[23]
*Elastase*	ACACGGTCAAGGAGATCAAC CGCTGGTGTTGGCCAGCAGG	540	[24]
*Lipase*	ATCTTCTCCGACTGGTTCGG CCGTGCCAGGACTGGGTCTT	383	[24]
*Aerolysin*	CAAGGAGGTCTGTGGCGACA TTTCACCGGTAGCAGGATTG	209	[25]
*intI1*	ACGAGCGCAAGGTTTCGGT GAAAGGTCTGGTCATACATG	565	[26]
*intI2*	GTGCAACGCATTTTGCAGG CAACGGAGTCATGCAGATG	403	[26]
Gene cassette(s) of class 1 integron	GGCATACAAGCAGCAAGC AAGCAGACTTGACCTGAT	Variable	[27]

**Table 2 antibiotics-08-00198-t002:** The prevalence of *Aeromonas* spp. in different aquatic animals and water.

*Aeromonas* spp.	Sources					
	Carp	Rainbow Trout	Sturgeon	Aquarium Fish	Crayfish	Water Samples
*A. hydrophila* (n = 49)	28	1	16	2	2	-
*A. veronii bv. sobria* (n = 14)	4	2	-	2	-	6
*A. bestiarum/piscicola* (n = 5)	4	-	-	1	-	-
*A. media* (n = 4)	1	2	-	-	-	1
*A. jandaei* (n = 1)	-	-	-	1	-	-
*A. aquariorum* (n = 1)	-	-	-	-	-	1

**Table 3 antibiotics-08-00198-t003:** The size and contents of gene cassettes and antibiotic resistance profile of sequenced MDR *Aeromonas* spp. isolates.

*Aeromonas* spp. (Numbers)	Fish Species/Water Sample	Cassette Size (kbp)	Gene Cassettes	Resistance Phenotype
*A. hydrophila* (2)	Sturgeon	1.0	*dfrA12-aadA2*	SXT, V, AMP, RD, NOR, OFL, TE, CIP
*A. hydrophila* (2)	Sturgeon	1.8	*dfrA12-orfF-aadA2*	SXT, V, AMP, RD, NOR, TE, CIP, DO, C
*A. hydrophila* (1)	Sturgeon	2.3	*aac(6’)-Ib-cr-arr3-dfrA27*	SXT, V AMP, RD
*A. hydrophila* (1)	Crayfish	2.0	*dfrB4-catB3-aadA1*	SXT, AMP, RD, TE, S, G, C
*A. veronii bv. sobria* (1)	Rainbow trout	2.3	*aac(6’)-Ib-cr-arr3-dfrA27*	STX, V, RD, TE, CIP S, G
*A. veronii bv. sobria* (1)	Water sample	0.7	*dfrA15*	STX, V, AMP, TE,
*A. media* (2)	Rainbow trout	1.7	*dfrA1-aadA1*	STX, V, RD, TE, DO, C
*A. aquariorum* (1)	Water sample	1.5	*dfrA1-orfC*	STX, V, AMP, RD, TE, DO, C

MDR, multidrug-resistant; SXT, trimethoprim/sulfamethoxazole; V, vancomycin; AMP, ampicillin; RD, rifampin; NOR, norfloxacin; OFL, ofloxacin; TE, tetracycline; CIP, ciprofloxacin, S, streptomycin; G, gentamycin; C, chloramphenicol.

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
