# Peer review of "Antimicrobial Resistance and Incidence of Integrons in *Aeromonas* Species Isolated from Diseased Freshwater Animals and Water Samples in Iran"

_antibiotics, 2019, doi:10.3390/antibiotics8040198_

Round 1
Reviewer 1 Report
In this study the authors demonstrated that freshwater species and water samples in Irán can be a source of Aeromonas species containing integron-mediated multiple drug-resistant phenotypes. Although the topics can be interesting, the originality and significance of the scientific content is low.
Here there are some of the criticisms:
The sequencing of the 16S rDNA is not discriminative enough to the identification of Aeromonas species. A MLSA analysis employing also other genes as gyrB, rpo... must be utilized. The authors must include a table showing the gene cassettes detected in the different Aeromonas species and their relationship with the phenotypic pattern of drug resistance. The figure 1 legend must be more explanatory The manuscript needs a english editing. Note that in the paragraph from lines 70 to 78 there are two sentences repeated.Author Response
Reviewer 1:
Reviewers' comment (1): The sequencing of the 16S rDNA is not discriminative enough to the identification of Aeromonas species.
Answer: thank you for the comment. Based on previous studies (for example, in the doi.org/10.1155/2013/127570), the use of biochemical tests and API system and confirmation by the 16S rRNA can be sufficient to detection of bacterial genus and species. In addition, in our study the final conformation was performed using the PCR of 16S rDNA, elastase, lipase and aerolysin
Reviewers' comment (2): The authors must include a table showing the gene cassettes detected in the different Aeromonas species and their relationship with the phenotypic pattern of drug resistance.
Answer: Table containing phenotypic and genotypic data of resistance pattern was added to the article.
Reviewers' comment (3): The figure 1 legend must be more explanatory
Answer: figure 1 legend was corrected.
Reviewers' comment (4): Note that in the paragraph from lines 70 to 78 there are two sentences repeated.
Answer: It was corrected.
Reviewer 2 Report
In this study, the authors showed that freshwater species can be a source of Aeromonas species containing integron-mediated multiple drug-resistant phenotypes. The cooperation of multi-drug resistance with integrons heightens the danger of co-selection and persistence of other resistance determinants under the selective pressure imposed by the use of antimicrobial agents. Although some other similar reports have confirmed the resistance to variety of antibiotics in Aeromonas strains, this study demonstrated that freshwater animals in Iran can be a source of multiply drug-resistant Aeromonas spp., and that antimicrobial resistance of animal origin be placed under increased surveillance in the future and that the use of antimicrobials in aquaculture be limited. The data presented in the current manuscript is very limited. I have a few comments for further improvement.
1) In antibiotic susceptibility test, please indicate the concentration of each antibiotic used, not just the total amount.
2) Please show the detail data about Aeromonas species identification, such as the cytochrome oxidase, catalase activity, and nitrate reduction tests.
3) What does A-G indicate in figure 1? Please give more details and analysis.
4) Please give more details about “The antimicrobial vulnerability of Aeromonas spp.” (Line 120), such as statistical analysis.
Author Response
Reviewer 2:
Reviewers' comment (1): In antibiotic susceptibility test, please indicate the concentration of each antibiotic used, not just the total amount.
Answer: The expressed antibiotic amount in the article is based on the concentration per disc. Since one disk is used for each test, the used concentration is equal to the total concentration.Reviewers' comment (2): Please show the detail data about Aeromonas species identification, such as the cytochrome oxidase, catalase activity, and nitrate reduction tests.
Answer: the results of mentioned tests were added to the article.Reviewers' comment (3): What does A-G indicate in figure 1? Please give more details and analysis.
Answer: It was corrected.Reviewers' comment (4): Please give more details about “The antimicrobial vulnerability of Aeromonas spp.” (Line 120), such as statistical analysis.
- Answer: Statistical Analysis was added.
Reviewer 3 Report
The work reports the distributions of class 1 and class 2 integrons in 74 Aeromonas species collected from “diseased” fish and water samples. The authors isolated the bacteria by culturing the samples on Agar and ran PCR (to detect integrase genes) and sequenced the PCR products (to detect the genes in the cassette array). The workflow is straightforward and the interpretation of the results are reasonable. However, manuscript lacks the novelty (several similar works have been done on another species) and work wasn’t conducted in-depth to provide an overall picture whether the patterns of integrons authors found are common in “all” Aeromonas species found in Aquaculture irrespective of their health status. The work can be improved significantly. I have a few concerns before the paper can be accepted for publication.
Major points:
Authors didn’t clarify what criteria was used to determine whether a fish was “diseased” or healthy? Unless authors present the integron distribution in Aeromonads isolates collected from healthy fish, it is not correct to say “Integron patterns” that authors found is only seen in Aeromonas species from “diseased” fish samples. The presentation of the result is poor. It is not clear whether the obtained integron pattern is largely found in A. hydrophia or common in all types of Aeromonas species from different fish species and water samples. Authors didn’t show any comparison on the integrons pattern found between 5 different fish species, and between fish and water samples. Similar results should be presented for antibiotic resistance pattern and conduct a statistical test to comment on the difference seen.Minor points:
1. Line 68. Most of the references haven’t been found in the reference list. Figure 1. A-G -> please clarify what are these seven groups of class 1 integrons? Add that info somewhere in the manuscript. Line: 13: Were all investigated 74-isolates were multi-drug resistant? If so, please clarify that in section 3.3. Line 25. “Optional pathogens” - please rephrase the term. Line 32. resistant -> resistance Line 36. “The most common….detected” – this sentence doesn’t make sense. Line 49-50. Aeromonads were also isolated from water so please mention that. Line 70. “Resistance test” -> “Susceptibility test” Line 103. Table 2. Row 3. (n=5) don’t match the sources of samples (n=4) Line 120. “vulnerability” -> “susceptibility” Line 133. “my country” -> Iran. Line 146. “The” -> “the”
Author Response
Reviewer 3:
Reviewers' comment (1): Authors didn’t clarify what criteria was used to determine whether a fish was “diseased” or healthy?
- Answer: the inclusion criteria for diseased fish were dermal ulceration, external and internal body haemorrhages, fins rot, lethargy, unbalanced swimming, and loss of appetite. The corresponding part in the "Materials and Methods" section was corrected.
Reviewers' comment (2): Unless authors present the integron distribution in Aeromonas isolates collected from healthy fish, it is not correct to say “Integron patterns” that authors found is only seen in Aeromonas species from “diseased” fish samples.
- Answer: In the present study, sampling was performed from diseased fish and environment. Furthermore, bacterial integron patterns are related to these sources.
Reviewers' comment (3): It is not clear whether the obtained integron pattern is largely found in A. hydrophia or common in all types of Aeromonas species from different fish species and water samples.
- Answer: As mentioned in the first line of section 3-2, the integrons presence and subsequently integrons genes was investigated in all 74 bacterial isolated and was not limited to Aeromonas bacteria.
Reviewers' comment (4): Authors didn’t show any comparison on the integrons pattern found between 5 different fish species, and between fish and water samples. Similar results should be presented for antibiotic resistance pattern and conduct a statistical test to comment on the difference seen.
- Answer: Based on the performed statistical analysis (Chi-square test), there was no significant difference among different bacterial species in the presence of integrons (p > 0.05). Additionally, the highest frequencies of class I integron were respectively related to sturgeon (n=5), rainbow trout (n=3), water samples (n=2) and crayfish (n=1). It should be mentioned that class I integron were not found in the carp and aquarium fish isolates.
Reviewers' comment (5): Minor points:
- Answer: All requested corrections were done.
Round 2
Reviewer 1 Report
The revised manuscript was slightly improved. I recommend some minor changes:
line 43: change "the main pathogens" by "important pathogens" lines 69-70. This sentence must be modified because these biochemical tests are also typical of Psudomonas or Vibrio species lines 108: change "were from" by "belonged to" Line 186: change "from a sturgeon fish" by "from a sturgeon fish was detected". line 188 inserte "was found" prior to A. hydrophila.Author Response
Reviewer 1:
Reviewers' comment (1): line 43: change "the main pathogens" by "important pathogens.
Answer: It was corrected.Reviewers' comment (2): lines 69-70. This sentence must be modified because these biochemical tests are also typical of Psudomonas or Vibrio species.
Answer: It was corrected.Reviewers' comment (3): lines 108: change "were from" by "belonged to".
Answer: It was corrected.Reviewers' comment (4): Line 186: change "from a sturgeon fish" by "from a sturgeon fish was detected".
Answer: It was corrected.Reviewers' comment (5): line 188 inserte "was found" prior to A. hydrophila.".
Answer: It was corrected.Reviewer 2 Report
The authors have addressed all my comments.
Author Response
Reviewer 2:
Reviewers' comment: Please show the detail data about Aeromonas species identification, such as the cytochrome oxidase, catalase activity, and nitrate reduction tests.
Answer: the results of mentioned tests were added to the article.